# Visual Question Answering Driven Eye Tracking Paradigm for Identifying Children with Autism Spectrum Disorder

## ABSTRACT

As a non-contact method, eye-tracking data can be used to diagnose people with Autism Spectrum Disorder (ASD) by comparing the differences of eye movements between ASD and healthy people. However, existing works mainly employ a simple free-viewing paradigm or visual search paradigm with restricted or unnatural stimuli to collect the gaze patterns of adults or children with an average age of 6-to-8 years, hindering the early diagnosis and intervention of preschool children with ASD. In this paper, we propose a novel method for identifying children with ASD in three unique features: First, we design a novel eye-tracking paradigm that records Visual Question Answering (VQA) driven gaze patterns in complex natural scenes as a powerful guide for differentiating children with ASD. Second, we contribute a carefully designed dataset, named VQA4ASD, for collecting VQA-driven eye-tracking data from 2-to-6-year-old ASD and healthy children. To the best of our knowledge, this is the first dataset focusing on the early diagnosis of preschool children, which could facilitate the community to understand and explore the visual behaviors of ASD children; Third, we further develop a VQA-guided cooperative ASD screening network (VQA-CASN), in which both task-agnostic and task-specific visual scanpaths are explored simultaneously for ASD screening. Extensive experiments demonstrate that the proposed VQA-CASN achieves competitive performance with the proposed VQA-driven eye-tracking paradigm. The code and dataset will be publicly available.

## CCS CONCEPTS

• **Information systems** → **Multimedia databases**; • **Applied computing** → **Health informatics**; *Psychology*.

## KEYWORDS

Eye movement, Autism Spectrum Disorder, Dataset, Visual Question Answer, Visual attention

## 1 INTRODUCTION

Autism Spectrum Disorder (ASD) is a disorder of very early brain development. Current clinical diagnosis methods such as Social

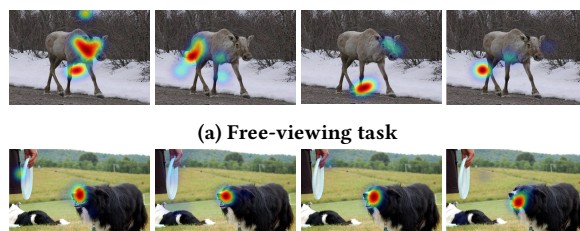

**(a) Free-viewing task**

**(b) VQA task (Q: Is the dog looking at the frisbee in the hand?)**

**Figure 1: The fixation maps of healthy people in two paradigms. In (a) free-viewing task, due to the absence of specific guidance, the diversity of the eye movement patterns is evident, where different subjects prefer to observe different parts of the cow. Instead, in (b) VQA task, guided by the explicit question, healthy subjects invariably watch the dog's eyes and the frisbee, which could reduce the intra-variance and benefit the ASD screening.**

Responsiveness Scale-2 (SRS-2) [5] and Repetitive Behavior Scale-Revised (RBS-R) [4] all depend greatly upon symptomatic observations and subjective judgments from clinicians. Therefore, developing automatic and objective ASD screening methods is an urgent need for the early diagnosis of ASD people.

It is evidenced that the eye movement patterns of people with ASD are significantly different from those of healthy people, thus revealing the potential of applying eye-tracking techniques to subjectively diagnose people with ASD [12, 14, 15, 28–31, 38]. Inspired by this, many works [6, 11, 26, 34, 39] try to analyze the eye-tracking data using machine-learning-based methods for diagnosis. Recently, benefiting from the strong learning ability of deep-learning technologies, some methods adopt deep neural networks [1, 7, 22, 40] and achieve promising results in diagnosing ASD.

However, early eye-tracking-based methods [2, 35, 41] usually utilize a simple visual search paradigm for identifying ASD, where only restricted or unnatural stimuli (*e.g.*, faces or letters) in isolation or even stimuli with low-level features are used for analysis. Recent studies [7, 22] attempt to develop new paradigms under complex scenes with natural background to analyze the difference between ASD and healthy people, but they mainly rely on the simple free-viewing paradigm, such as watching images or videos without any specific purpose. Due to the lack of explicit guidance in the free-viewing style, the eye movements are susceptible to individual preference, leading to large intra-variance (*i.e.*, the eye movements of different healthy people are highly dissimilar when viewing the same image). As shown in Fig. 1a, the gaze patterns of different healthy people are significantly diverse when watching the same image, making the ASD classification quite challenging. Besides, since collecting eye-tracking data from preschool children is much more difficult, the eye-tracking-based datasets are mainly

taken from adults [3, 20, 22] or children with an average age of 6-to-8 years [12, 16], which highly limits the early diagnosis and intervention of young children.

To tackle the above issues, in this paper, we propose a novel eye-tracking paradigm in which the Visual Question Answering (VQA) task is designed to collect distinctive eye movement patterns, which provides a better understanding of how visual attention is deployed in the brain of children with ASD in real-life behaviors. In our eye-tracking experiments, the images combined with the corresponding questions are shuffled and presented in consecutive trials. The subjects are required to search the related regions of the images to answer the given questions. Introducing the VQA-driven eye-tracking paradigm provides powerful guidance to get the subject concentrated, which can alleviate the effect of large intra-variance in the free-viewing paradigm. As shown in Fig. 1b, with the guidance of the VQA task, the gaze patterns of different healthy people are highly similar, which can benefit the following identification. Moreover, the VQA task requires the subjects to interact and reason for answering the questions, and thus can partially reflect the interaction ability of the subjects, which also helps the accurate prediction of ASD classification.

To evaluate the effectiveness of the VQA paradigm, we launch recruitment for preschool children diagnosed with and without ASD to voluntarily take part in our experiments. So far, we have successfully collected effective eye movements from 42 ASD and 26 typically developed (TD) children (control group). There are no statistical differences between the two groups in terms of visual acuity, age, etc. To our knowledge, our collected dataset, named VQA4ASD, is the first eye movement dataset focusing on 2-6-year-old children, allowing the community a better comprehension of the mechanism of how visual attention influences ASD's behaviors.

Furthermore, built upon the collected dataset, we propose a novel model, VQA-guided cooperative ASD screening network (VQA-CASN), which consists of two branches to identify ASD and healthy people jointly. In the proposed VQA-CASN, one branch captures the eye movement features in a task-agnostic manner, where the differences in gaze patterns across the whole image are utilized for classification. The other branch only focuses on the differences in the regions of interest, *i.e.*, question-related regions, aiming at reducing the variance of TD group to better differentiate ASD and healthy people in a task-specific manner. These two branches jointly learn the subject's attention under the VQA task, which provides a more efficient solution in real clinical scenarios. We conduct a series of experiments on our collected dataset and achieve state-of-the-art performance.

Overall, this paper has four main contributions:

- We design a novel eye-tracking paradigm where Visual Question Answering (VQA) driven gaze patterns are employed for ASD screening. The proposed paradigm can not only mitigate the large intra-variance in the previous free-viewing paradigm but also assess the interaction ability of ASD children through visual patterns.
- We construct a delicately designed dataset, named VQA4ASD, for collecting VQA-driven eye-tracking data from 2-6-year-old ASD and TD children. To our knowledge, this is the first dataset focusing on preschool children in the vision-based

ASD intervention community, which could benefit the early objective diagnosis greatly.
- We further develop a VQA-guided cooperative ASD screening network (VQA-CASN) where the task-agnostic and task-specific branches jointly learn the gaze patterns for better ASD screening.
- We conduct comprehensive experiments on our collected dataset. Results show that our method achieves state-of-the-art performance, demonstrating the effectiveness of our proposed paradigm and designed model for ASD screening.

## 2 RELATED WORK

### 2.1 Eye Tracking Paradigm

Current ASD diagnosis is usually performed by subjective and manual screening methods. However, these subjective measurements are both time-consuming and clinically demanding. Since it has been verified that the visual attention of people with ASD presents distinctive characteristics from healthy people, recent studies [25, 26, 36, 39] aim at developing objective and automatic screening methods to distinguish the ASD individuals, where the subjects are asked to freely view a set of images or videos and an eye tracker is used to collect the subjects' eye movements.

In order to collect distinctive gaze patterns for identifying ASD precisely, various paradigms have been explored. Gong *et al.* [16] use dynamic images with repetitive movements. Hochhauser *et al.* [20] utilize changed blindness paradigm and social films. Chita-Tegmark *et al.* [8] present visual and auditory stimuli in pairs. Au-Yeung *et al.* [3] display the paired visual stimuli simultaneously. Some works [27, 33] use the visual search paradigm with simple stimuli to ask the subjects to locate the predefined objects. Recently, [7, 22] develop image free-viewing (IV) paradigm, where the subjects watch natural images or videos. Such free-viewing paradigm cannot provide useful guidance, leading to a large variance among healthy people. In contrast, our VQA-driven task can alleviate the negative effect of intra-variance under explicit guidance.

The visual search paradigm is most related to our VQA paradigm, but our work is superior to the previous visual search paradigm in three aspects: 1) Previous visual paradigms mainly use restricted or unnatural stimuli (*e.g.*, faces or letters) in isolation or even stimuli with only low-level features. Our VQA paradigm explores visual attention with more natural stimuli (*e.g.*, complex scenes taken with a natural background), providing a better understanding of how attention is deployed in children with ASD when viewing the real world. 2) Our VQA paradigm designs more kinds of questions (*e.g.*, how many, is it, where, what) than visual search (count), which can reveal the impairments of both communication and social interaction in ASD children. More importantly, the VQA task has been fully studied in the multimedia field, so abundant datasets and models can bring new insights about visual saliency for ASD screening. 3) Most previous visual search paradigms published in medical-related articles did not release the dataset. So far, only [12] has released an image-viewing dataset collected from school children and adults. In this work, we collect the first VQA-driven dataset focusing on identifying preschool children with ASD, which is not explored in most previous works due to the difficulty of data collection. We

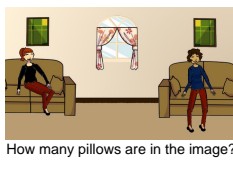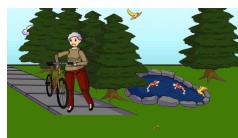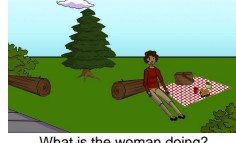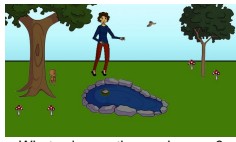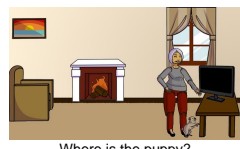

How many pillows are in the image?    Is it the old lady holding the bike?    What is the woman doing?    What color are the mushrooms?    Where is the puppy?

**(a) The examples of abstract scene images and the corresponding questions.**

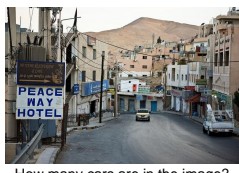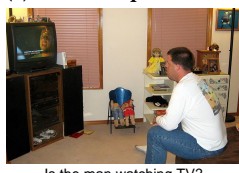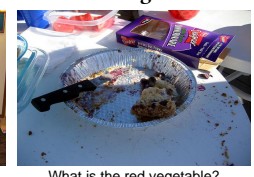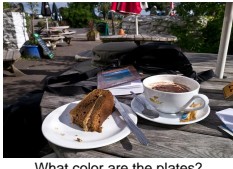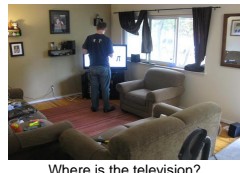

How many cars are in the image?    Is the man watching TV?    What is the red vegetable?    What color are the plates?    Where is the television?

**(b) The examples of real scene images and the corresponding questions.**

**Figure 2: The examples of VQA stimuli and the corresponding questions in our VQA4ASD.**

believe that our dataset can largely enrich the studies of automatic computer-aided diagnosis and multimedia applications.

## 2.2 ASD Screening Methods

To analyze the collected eye-tracking data, some works introduce machine learning-based methods [26, 37, 39] to distinguish ASD. Thanks to the powerful representation abilities of deep learning techniques, deep learning-based methods have achieved significant performance in ASD screening. Jiang and Zhao [22] propose the first deep neural network to identify ASD. Chen and Zhao [7] present to use multi-modality (*i.e.*, photo-taking and image-viewing) in ASD screening and achieve state-of-the-art performance.

Compared with previous works which usually adopt a single-branch architecture, our proposed dual-branch network has an additional task-specific branch that focuses on analyzing the gaze patterns in question-related regions. Benefiting from the explicit guidance of the VQA task, adding the task-specific branch can reduce the large variance of the gaze patterns within the TD group, leading to a better diagnosis performance.

## 3 DATASET

In this section, we construct a VQA-driven eye-tracking dataset, named VQA4ASD, for the early diagnosis of ASD children with two objectives: (1) to reduce the intra-variance in the control group by introducing explicit guidance; (2) to reveal the social and communication skills, especially the ability to receive questions and make corresponding reactions, which indicates the social and attentional deficits of subjects. Compared with the existing datasets [7, 22], our VQA4ASD provides the first VQA-driven eye movement data collected from preschool children, ranging from 2 to 6 years, which can benefit the early diagnosis of preschool children with ASD. Some VQA stimuli and the corresponding questions of ASD and control group are provided in Fig. 2. More examples are given in the supplementary material.

## 3.1 Visual Stimuli

To construct our VQA4ASD dataset for diagnosing ASD children, we choose the images and the corresponding questions from VQA

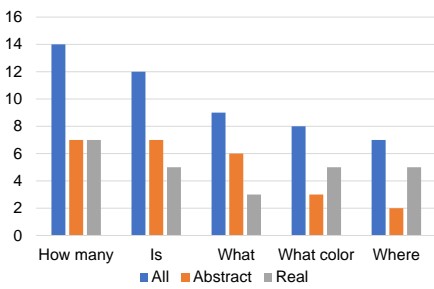

**Figure 3: The statistics of the types of questions.**

v2.0 dataset [17], where 25 images are selected from real scenarios and 25 images are selected from abstract scenarios. To guarantee that the images contain a large variety of scenes, we carefully select the images covering common scenarios of daily life for children. Specifically, the abstract scene images include 1 indoor image with single person, 2 outdoor images with single person, 2 indoor images with multiple persons, 6 outdoor images with multiple persons, 7 indoor images with people and animals, and 7 outdoor images with people and animals. The real scene images include 1 image with natural scene, 1 image with single person, 2 images with person and animals, 3 images with multiple persons, 5 images with single person and objects, 5 images with animals, and 8 images with buildings or objects. Additionally, we also arrange an image free-viewing task accompanied by VQA task for a fair comparison with free-viewing paradigm (see details in Section 5.1).

Besides, since children with ASD may show a center bias [12, 39] and some questions of VQA may be difficult for preschool children, all of the questions we choose are relatively simple and most of the answers to the questions cannot be directly found in the center of the screen. The distribution of question types is shown in Fig. 3. We can see that the types of questions are abundant, including "How much", "Is this", "What", and "Where", to ensure that gaze patterns can be recorded in various scenarios.

## 3.2 Subjects

We collect eye-tracking data from 68 preschool children aged 2-to-6 years old, including 69 children with 42 ASD and 27 TD. All ASD

children are determined by an expert-level clinician and evaluated by Autism Behavior Checklist (ABC) [24] and Childhood Autism Rating Scale (CARS) [32], which can exclude unqualified children who may also suffer from other non-ASD developmental disabilities. Before the experiments, we obtained informed consent from children's parents. Besides, due to social interaction disorder, it is difficult for children with ASD to cooperate on finishing the experiment tasks and even calibration, and several TD children also fail to complete all experiments due to losing patience and interest. Therefore, only 24 children with ASD and 24 TD, whose ages are $4.0 \pm 1.2$ and $4.6 \pm 0.8$, complete all experiments.

### 3.3 Apparatus and Experimental Procedure

We use SensoMotoric Instruments (SMI) IView X RED eye tracker to record the subjects' eye movements by showing the stimuli on a displayer with $1280 \times 768$ resolution.

During training, we set the sampling frequency of the eye tracker to 120 Hz. In order to avoid the subjects from losing their concentration due to the long-time continuous experiment, we split VQA tasks into two parts, each consists of 25 images. Before each trial begins, a five-point calibration is attached to ensure the precision of the eye movement data. During each trial, a gray image is first displayed for three seconds and accompanied by the corresponding question, and then the image is presented for five seconds. We ask the question again during the image presentation to avoid the situation where children do not hear the question clearly. When we observe that children are distracted from the displayer, we will remind them to view the screen with a non-leading sentence (*e.g.*, "Let's look at the screen"). The duration of each subject finishing the VQA experiments is about 15 minutes.

### 3.4 Data Processing

For the obtained raw eye movement data, we need to further process the data for the training of the diagnosis model. In the raw data, the sequences of continuous fixation events separated by saccades or blinks are considered fixations. So we first generate the location of each fixation by averaging the continuous fixation events' horizontal and vertical positions. The duration of the averaged fixation is calculated as the time from the first fixation event of the fixation sequence to the first non-fixation event after that. Since the subject's first fixation is often the last fixation position on the previous gray image, it is meaningless and cannot reflect the subject's interest. In order to eliminate such possible distractions, we discard the first fixation for all subjects. Finally, we generate eye movements that are rounded to integers and recorded for further training and testing.

It should be noted that all eye movement data in our dataset are recorded, including the visual stimuli, the corresponding questions, the raw data, as well as processed data, which will be made publicly available upon publication of the work.

### 3.5 Statistics of VQA4ASD

The comparison between the previous dataset Saliency4ASD [12] and our VQA4ASD are summarized in Table 1. We can see that our collected VQA4ASD involves more individuals and younger

children. Besides, VQA4ASD includes both VQA and image free-viewing (IV) paradigms, which facilities the fair comparison between different paradigms.

**Table 1: The advantages of VQA4ASD over Saliency4ASD [12].**

|  | Saliency4ASD [12] | VQA4ASD (ours) |
|---|---|---|
| #Subjects | 28 (14 ASD & 14 TD) | 48 (24 ASD & 24 TD) |
| Age range | 5-12 (avg. 8) | 2-6 (avg. 4) |
| Paradigm | IV | VQA & IV |

## 4 METHOD

### 4.1 Overview

The overview of our proposed VQA-guided cooperative ASD screening network (VQA-CASN) is shown in Fig. 4. The intuition behind our design is to cooperate with the task-agnostic (the gaze patterns across the whole image) and task-specific (the gaze patterns within the question-related region), which can largely reduce the intra-variance of TD group and further leverage the identification performance under the guidance of VQA task.

Specifically, VQA-CASN contains three modules, *i.e.*, image feature extraction, fixation sequence feature extraction, and classification. The first component of our model is extracting feature representations of images, which is shared between the following two branches. Then, in the fixation sequence feature extraction module, we design a dual-branch architecture for extracting both task-agnostic and task-specific features of the fixation sequences. The task-agnostic branch focuses on extracting features from the whole fixation sequence. The task-specific branch attempts to obtain local features from VQA task-related regions. Finally, after fusing the two above features and the location and duration of fixations, we feed the concatenated features to a fully connected (FC) layer with a sigmoid function to compute the probability of suffering from ASD in the final classification module by weighting the contributions of different gaze patterns. Benefiting from the cooperation of the above modules, VQA-CASN can simultaneously learn the difference of gaze patterns on both task-agnostic and question-related regions, where two branches are complementary to each other and yield favorable accuracy.

### 4.2 Image Feature Extraction

In order to learn the effective image representation of the subject's scanpath, we use the popular feature extractor ResNet-50 [19] to learn the image features. For one image $x$, we extract a 2048-$d$ feature from the top convolutional layer of ResNet-50 as the image feature of each fixation point which is shared in the following two-branch module.

### 4.3 Fixation Sequence Feature Extraction

After obtaining the image feature of the image $x$, given the corresponding gaze pattern of one subject watching the image $x$, we further generate the feature of the fixation sequence by a dual-branch architecture, where two branches adopt the same network structure, *i.e.*, a variant [18] of Long Short Term Memory (LSTM)

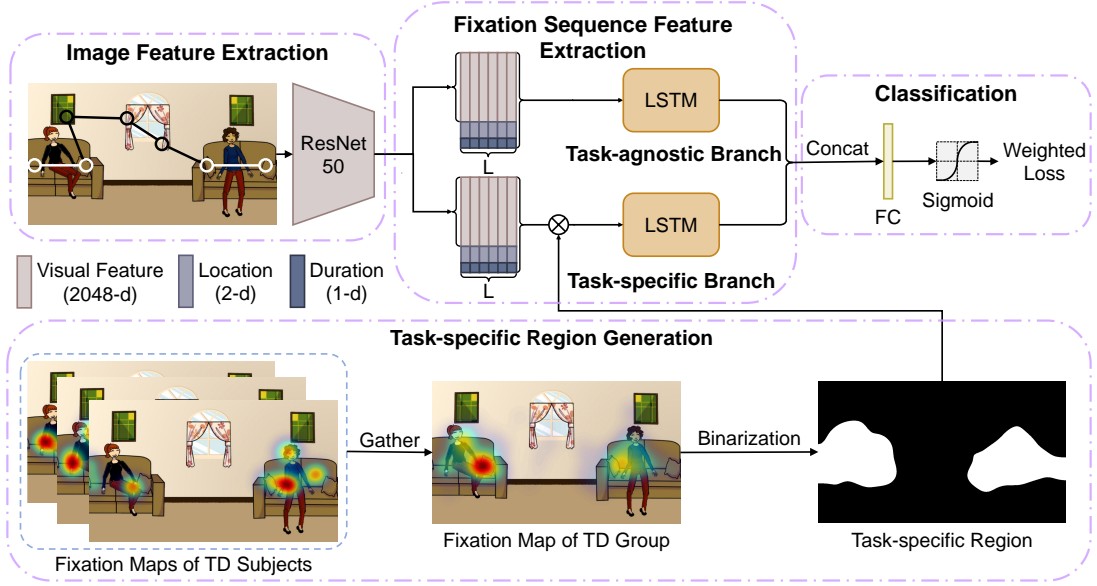

**Figure 4: The overview of our proposed VQA-CASN. We first use a ResNet-50 [19] to extract the image feature which is concatenated with the location and duration of the fixation point. Then, we design a dual-branch architecture to extract the feature for the fixation sequence. Specifically, the task-agnostic branch captures the feature across the whole image using LSTM [18] network, and the task-specific branch acquires the LSTM feature within the question-related region, which is generated according to the fixation map of the TD group. Finally, the two features are concatenated and fed to a fully connected (FC) layer with sigmoid function for identification.**

network [21], to encode the temporal information of the gaze pattern. Note that the two LSTMs do not share the parameters to capture abundant features for different purposes.

**Task-agnostic branch.** In order to analyze the difference between ASD and TD across the whole image, we first extract the feature of the fixation sequence in a task-agnostic manner. Specifically, we first extract the 2048-$d$ visual feature at each fixation and concatenate it with the location and duration as a 2051-$d$ feature of each fixation. Then we stack the feature of each location in temporal order to generate the feature of the fixation sequence. To reduce the interference of the redundant fixations, we manually control the maximum number of fixations as $L$ and discard the rest fixations. Consequently, we get the feature representation of the fixation sequence denoted as $F \in \mathbb{R}^{L \times 2051}$.

To obtain more powerful features, we further feed $F$ into the LSTM [18] in the task-agnostic branch to capture the temporal information by updating the hidden state repeatedly. We treat the final hidden state as the task-agnostic feature $S_{ta}$ of one gaze pattern as follows:

$$S_{ta} = \text{LSTM}_\phi(F), \tag{1}$$

where $\phi$ is the parameters of LSTM in task-agnostic branch.

**Task-specific region generation.** Thanks to the VQA task, except for the task-agnostic feature, we can pay more attention to the task-specific regions to further differentiate the gaze patterns between ASD and healthy people. Before generating the task-specific features, we should know where the regions are related to the question and answer. We achieve this goal by regarding the fixation maps of the control group as possible question-related regions. This

strategy is reasonable since our VQA task is performed in an interactive Q&A manner, which is much more attractive to healthy children, so they are very likely to cooperate in the collection process, which guarantees that the TD group's fixation maps are highly relevant with the corresponding questions and answers. We have empirically validated this by comparing the results with manually labeling the task-relevant regions. The results are relatively close (see Sec. 5.4 for details). Considering that manual labeling requires extra annotations, we therefore design the task-specific branch by averaging all TD children's fixation maps, making our model easily extendable to other new VQA stimuli because it does not require any additional annotations.

Specifically, we first gather the fixation points of the control group by setting the location of fixation points to 1 and other pixels to 0. Then we follow [12] to obtain the averaged fixation map $M$ of the control group by applying the Gaussian smoothing and min-max normalization. Subsequently, we can get the task-specific region $R$ by using binarization on the averaged fixation map.

**Task-specific branch.** The task-specific branch focuses on the gaze patterns within the question-related regions, so we first obtain the task-specific feature of the fixation sequence, denoted as $F_{ts} \in \mathbb{R}^{L \times 2051}$, with the guidance of the task-specific region $R$ by replacing the feature that falls outside the question-related region with 0.

Similar to the task-agnostic branch, the task-specific feature of the fixation sequence $F_{ts}$ is then forwarded into LSTM [18] in the task-specific branch to capture the question-related feature of the gaze pattern $S_{ts}$:

$$S_{ts} = \text{LSTM}_\varphi(F_{ts}), \tag{2}$$

where $\varphi$ is the parameters of LSTM in task-specific branch.

## 4.4 Classification

Once the $S_{ta}$ and $S_{ts}$ have been obtained, they are concatenated and forwarded to a multilayer perceptron (MLP), *i.e.*, an FC layer with sigmoid function, to acquire the predicted score $\hat{y}$ which can estimate whether the gaze pattern belongs to a subject with ASD or not.

$$S = \text{Concat}(S_{ta}, S_{ts})$$
$$\hat{y} = \text{MLP}_\theta(S), \tag{3}$$

where $\theta$ is the parameters of the MLP layer.

As different visual stimuli show distinct abilities for diagnosing ASD, we design an adaptive weighting loss to better differentiate ASD and TD. Specifically, for each visual stimulus, we calculate the cosine similarity of the corresponding fixation maps of ASD and TD groups to define a weighted cross entropy loss:

$$w = \text{softmax}(1 - \cos(\bar{M}^{ASD}, \bar{M}^{TD})), \tag{4}$$

$$\mathcal{L}(y, \hat{y}) = -w \left( y \log(\hat{y}) + (1 - y) \log(1 - \hat{y}) \right), \tag{5}$$

where $M$ is the fixation map of the subject for single visual stimulus. $\bar{M}^{ASD}$, $\bar{M}^{TD}$ represent the averaged fixation map of ASD group and TD group for single visual stimulus, respectively. $y \in \{0, 1\}$ and $\hat{y} \in \{0, 1\}$ denote the ground-truth label and the predicted score (1 is ASD and 0 is TD).

In the inference, given one subject, we generate the final subject's predicted score by averaging the predictions obtained from all stimuli using (4) as a weighting function, and a pre-defined threshold of 0.5 is used to identify ASD.

## 5 EXPERIMENTAL RESULTS

### 5.1 Experiment Settings

**Dataset.** We conduct all experiments on our VQA4ASD dataset which is divided into two parts, the abstract and real scenes, named VA and VR, respectively. Besides, in order to give a comparison with previous approaches [7] and facilitate future works, we also arrange an additional image free-viewing (IV) paradigm to collect the eye movements while constructing VQA4ASD dataset. We set the ratio of the number of IV images to VQA images to 1:1 and conduct experiments on the same subjects who complete all tasks. The dataset consists of a total of 4800 samples, with 2400 samples for IV and VQA, respectively. The details of three kinds of paradigms are as follows:

- **IV** (Image free-viewing, abbreviated as IV) is collected by asking the subjects to freely view images without guidance. It contains the most distinctive top 50 images out of a total of 300 images according to a Fisher-score-based image selection strategy [7, 22] from Saliency4ASD dataset [12].
- **VA** (Abstract scene in VQA4ASD, abbreviated as VA) includes 25 clipart-style images selected from the abstract scenes in VQA v2.0 dataset [17].
- **VR** (Real scene in VQA4ASD, abbreviated as VR) includes 25 images selected from the real scenes in VQA v2.0 dataset [17]. The details of VA and VR have been given in Sec. 3.

**Training details.** All images are resized to $800 \times 600$ and normalized (*i.e.*, make the mean and standard deviation of the images 0 and 1). The parameters of ResNet-50 [19] are initialized using pre-trained parameters on ImageNet [10]. The size of the hidden

state of LSTM [18] is 512. The length of the fixation sequence $L$ is 14. During the training stage, we set the batch size to 10. All models are trained for 10 epochs using the Adam [23] optimizer with the learning rate of $10^{-3}$ and the weight decay of $10^{-5}$. The gradient clip strategy is applied by keeping the gradient no larger than 10.

**Evaluation metrics.** Following previous work [7], we use the leave-one-subject-out method for cross-validation and evaluate the models with four metrics, including accuracy (Acc), sensitivity (Sen), specificity (Spec), and area under the ROC curve (AUC). In addition to evaluating the subject-level performance of each subject, we also evaluate the the scanpath-level performance of each gaze pattern to validate which model can select more distinctive gaze patterns for diagnosis.

**Baseline methods.** We are the first to propose VQA-driven eye-tracking data for ASD diagnosis, and Jiang *et al.* [22] and Chen *et al.* [7] use the free image-viewing task, which are the most related works and thus are chosen as our compared methods. For fairness, we use the same hyper-parameters in [22] and [7] except the batch size of 10. Besides, we also implement a non-deep-learning simple baseline to verify the effectiveness of VQA paradigm, where the Fisher score method [13] is used to extract the gaze pattern's feature [7, 22] followed by a linear Support Vector Machine (SVM) classifier [9] for classification.

### 5.2 Effectiveness of VQA Paradigm

**Qualitative analysis.** As shown in Fig. 1, the guidance of the question effectively gather the TD subjects' attention on the dog's eyes. In contrast, TD subjects will view the different parts of the cow in the free-viewing task. Furthermore, as shown in Fig. 5, the visual patterns of TD subjects and ASD subjects show significant difference in VQA task. Since TD subjects can communicate and interact with us normally, their gaze patterns are highly related to the questions. In contrast, due to the social interaction disorder, ASD subjects do not respond to the questions and will view the images according to their personal consciousness.

**Quantitative analysis.** To show the superiority of the VQA paradigm, we make comprehensive comparisons between different paradigms. We perform experiments on VQA (VA+VR) and IV paradigms using four models, *i.e.*, Fisher score method, Jiang and Zhao [22], Chen and Zhao [7], and our proposed VQA-CASN. The comparisons are summarized in Table 2.

For the simple Fisher score method, we can see that the VQA paradigm achieves better performance than IV in all metrics. Considering that the 50 images of IV paradigm are chosen from 300 images based on the analysis of collected gaze patterns [7], our paradigm simply chooses 50 images from VQA dataset [17] without any guidance of gaze patterns, but still achieves better performance, which can suggest the effectiveness of the designed paradigm.

Besides, we compare two paradigms using the model proposed by Jiang and Zhao [22]. Compared with IV paradigm, our VQA paradigm gains better performance in terms of accuracy, sensitivity, and AUC in subject-level (*e.g.*, an AUC improvement of 20.3%). Noted that the model [22] fuses the scanpath features before prediction, so we do not evaluate the performance on scanpath level. We also compare two paradigms using the model proposed by Chen and Zhao [7]. Compared with IV paradigm, our VQA paradigm achieves

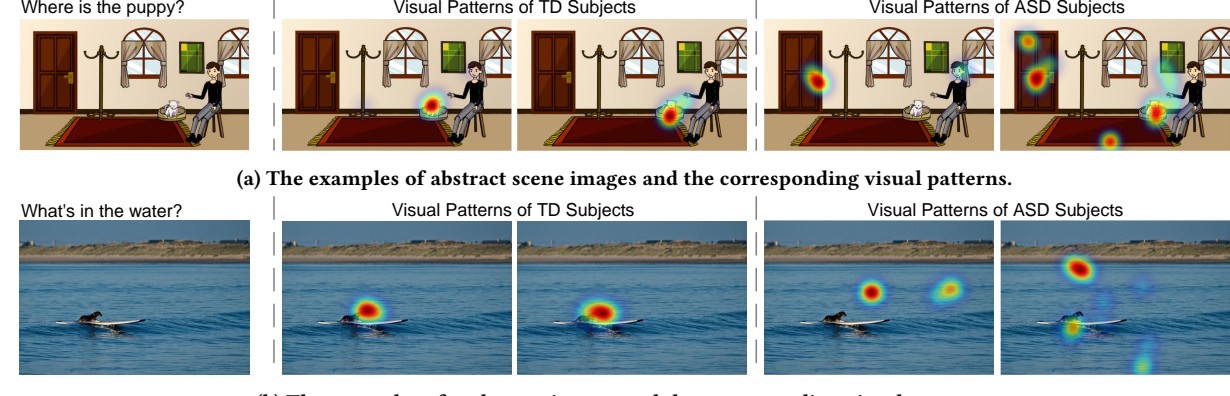

(a) The examples of abstract scene images and the corresponding visual patterns.

(b) The examples of real scene images and the corresponding visual patterns.

Figure 5: The examples of VQA stimuli and the corresponding visual patterns in VQA task.

Table 2: The comparison between different paradigms on four models, including Fisher score, Jiang and Zhao [22], Chen and Zhao [7] and our proposed VQA-CASN model.

| | | Metric | Paradigm | Acc | Sen | Spec | AUC |
|---|---|---|---|---|---|---|---|
| Fisher Score | Subject-level | | IV | 0.646 | 0.583 | 0.708 | 0.750 |
| | | | VA+VR | **0.792** | **0.750** | **0.833** | **0.884**+13.4% |
| | Scanpath-level | | IV | 0.623 | 0.606 | 0.638 | 0.660 |
| | | | VA+VR | **0.697** | **0.608** | **0.776** | **0.742**+8.2% |
| [22] | Subject-level | | IV | 0.771 | 0.708 | **0.833** | 0.682 |
| | | | VA+VR | **0.792** | **0.750** | **0.833** | **0.885**+20.3% |
| [7] | Subject-level | | IV | **0.938** | 0.917 | **0.958** | 0.977 |
| | | | VA+VR | **0.938** | **0.958** | 0.917 | **0.981**+0.4% |
| | Scanpath-level | | IV | 0.653 | 0.644 | 0.661 | 0.682 |
| | | | VA+VR | **0.744** | **0.719** | **0.768** | **0.807**+12.5% |
| VQA-CASN | Subject-level | | IV | 0.917 | **1.000** | 0.833 | 0.971 |
| | | | VA+VR | **0.958** | **1.000** | **0.917** | **0.991**+2.0% |
| | Scanpath-level | | IV | 0.792 | 0.858 | 0.726 | 0.837 |
| | | | VA+VR | **0.833** | **0.872** | **0.794** | **0.863**+2.6% |

better accuracy, sensitivity, and AUC score in subject-level and all metrics in scanpath-level performance (*e.g.*, an AUC improvement of 0.4% in subject-level and 12.5% in scanpath-level).

Furthermore, we compare two paradigms tested on our proposed VQA-CASN model. By introducing task-specific guidance, VQA paradigm can better employ helpful stimuli and question-related clues to benefit the identification, leading to a performance gain compared with IV paradigm (*e.g.*, an AUC improvement of 2.0% in subject-level and 2.6% in scanpath-level). By contrast, IV paradigm tested on VQA-CASN cannot obtain useful guidance from the TD group's fixation maps and results in the performance drop, especially in the subject-level performance compared with [7] (from 0.977 to 0.971 in AUC), which also verifies that free-viewing paradigm has the disadvantage of large variance in the TD group.

## 5.3 The Superiority of VQA-CASN

We conduct the comparison experiments between different models on VA, VR, and VA+VR and the results are summarized in Table 3. It should be noted that the result of VA+VR is supposed to be within the range of VA and VR's results since VA+VR is the average of VA and VR on the scanpath-level.

We first report the comparison for the performance on subject-level. For VA or VR, our model achieves the best accuracy, sensitivity, and AUC, and only a little drop in specificity. For VA+AR, we achieve the best results on all metrics. Furthermore, we also report the comparison for the prediction performance on scanpath-level. On VA, VR, and VA+VR, VQA-CASN shows significant improvements on all metrics (*e.g.*, For VA+AR, 8.9% in Acc, 15.3% in Sen, 2.6% in Spec, 5.6% in AUC), which clearly demonstrate that our VQA-CASN can select more distinctive gaze patterns for diagnosis.

## 5.4 Ablation Study

Our VQA-CASN proposes several strategies to boost performance. To verify their effectiveness, we conduct ablations to explore the role of each strategy, and the results are shown in Table 4.

**Effect of the dual branch.** As shown in the first and second rows of Table 4, removing either the task-specific or task-agnostic branch leads to a performance drop. In terms of AUC, removing the task-specific branch leads to a performance reduction of 1.7% and removing the task-agnostic branch also drops 0.7%. It is not surprising because the question-related guidance can help identify the abnormal visual attention of ASD children and the task-agnostic branch can also guarantee the model identifies healthy people more accurately. Besides, the task-specific branch is more effective than the task-agnostic branch, because the task-specific branch has a closer relation with the VQA paradigm and fully utilizes the additional information from the VQA paradigm.

**Effect of the weighting loss.** To figure out the impact of the weighted loss in VQA-CASN, we conduct an ablation study on VA+VR by removing the weighted loss. As shown in the third row of Table 4, our weighted loss can largely improve the performance of ASD identification (8.3% in accuracy, 12.5% in sensitivity, 4.2% in specificity, and 2.4% in AUC), due to the consideration of the

Table 3: The comparison results of Jiang *et al.* [22], Chen *et al.* [7] and our proposed VQA-CASN method. Since the method [22] fuses the features of scanpaths before prediction, we cannot evaluate the performance on scanpath level.

| Paradigm | #Images | Method | Subject-level | | | | Scanpath-level | | | |
|---|---|---|---|---|---|---|---|---|---|---|
| | | | Acc | Sen | Spec | AUC | Acc | Sen | Spec | AUC |
| VA | 25 | Jiang and Zhao [22] | 0.729 | 0.750 | 0.708 | 0.790 | - | - | - | - |
| | | Chen and Zhao [7] | 0.917 | 0.875 | **0.958** | 0.967 | 0.764 | 0.737 | 0.792 | 0.831 |
| | | VQA-CASN (Ours) | **0.938** | **0.958** | 0.917 | **0.974** | **0.832** | **0.855** | **0.808** | **0.867** |
| VR | 25 | Jiang and Zhao [22] | 0.750 | 0.708 | 0.792 | 0.797 | - | - | - | - |
| | | Chen and Zhao [7] | 0.896 | 0.875 | **0.917** | 0.964 | 0.723 | 0.702 | 0.745 | 0.780 |
| | | VQA-CASN (Ours) | **0.938** | **1.000** | 0.875 | **0.990** | **0.834** | **0.888** | **0.780** | **0.860** |
| VA+VR | 50 | Jiang and Zhao [22] | 0.792 | 0.750 | 0.833 | 0.885 | - | - | - | - |
| | | Chen and Zhao [7] | 0.938 | 0.958 | **0.917** | 0.981 | 0.744 | 0.719 | 0.768 | 0.807 |
| | | VQA-CASN (Ours) | **0.958** | **1.000** | 0.917 | **0.991** | **0.833** | **0.872** | **0.794** | **0.863** |

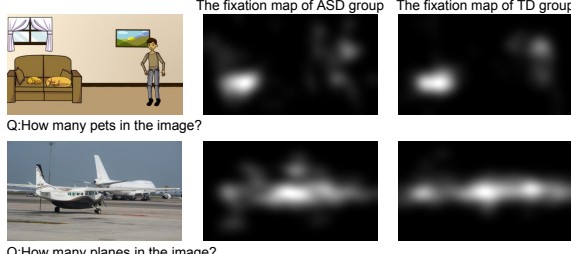

Q:How many pets in the image?

Q:How many planes in the image?

(a) The fixation maps of VQA samples with lower weights.

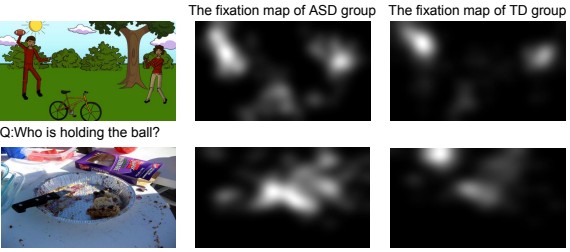

Q:Who is holding the ball?

Q:What is the red vegetable?

(b) The fixation maps of VQA samples with higher weights.

Figure 6: The visualizations of fixation maps on ASD and TD group.

Table 4: The ablation studies for VQA-CASN on VA+VR.

| Method | Acc | Sen | Spec | AUC |
|---|---|---|---|---|
| w/o task-specific branch | 0.958 | 1.000 | 0.917 | 0.974 |
| w/o task-agnostic branch | 0.958 | 1.000 | 0.917 | 0.984 |
| w/o weighting loss | 0.875 | 0.875 | 0.875 | 0.967 |
| w/o location and duration | 0.958 | 0.958 | 0.958 | 0.990 |
| labeling task-specific region manually | 0.917 | 0.917 | 0.917 | 0.988 |
| VQA-CASN | 0.958 | 1.000 | 0.917 | 0.991 |

different contributions of each visual stimuli. Besides, we show some images and corresponding fixation maps in Fig. 6, where the top two rows give examples with lower weights due to the similar

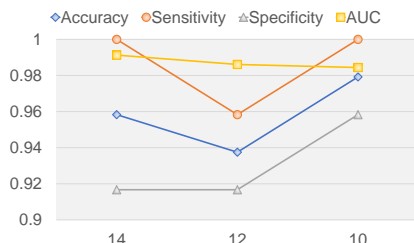

Figure 7: The results on different lengths of fixation sequence $L$.

fixation maps of ASD and TD group, and the bottom two rows are the images with higher weights with significant diverse fixation maps of ASD and TD group.

**Effect of the location and duration of fixation points.** To verify the effect of the location and duration of fixation points for generating distinctive representations, we remove the location and duration and the result is given in the fourth row of Table 4. We can see that adding location and duration is beneficial for improving performance in terms of Sen and AUC.

**Effect of using TD group's fixation map as the task-specific region.** To verify the effect of TD group's fixation map as the task-specific region, we use the manually labeled task-specific region and the result is given in the fifth row of Table 4. We can see that using manually labeled task-specific region can lead to a performance drop in terms of Acc, Sen and AUC (4.1%, 8.3% and 0.3%).

**Length of fixation sequence.** We investigate the effect of the length of fixation sequence $L$. As shown in Fig. 7, by setting $L$ to 14, we achieve the best performance in terms of AUC.

## 6 CONCLUSION

In this paper, we propose a novel VQA-driven paradigm for ealy diagnosis ASD children. We construct the first VQA4ASD dataset, which contains the data from 2-to-6-year-old preschool children. Moveover, we design a dual-branch VQA-CASN model to identify ASD and healthy individuals. The results show the effectiveness of our proposed VQA paradigm and the VQA-CASN model.

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
