# OpenReview forum: "Visual Question Answering Driven Eye Tracking Paradigm for Identifying Children with Autism Spectrum Disorder"
_acmmm.org/ACMMM/2024/Conference — MM2024 Poster_

### Official Review · Reviewer_vDQi · 2024-05-08

**Rating:** 3
**Confidence:** 4

**Summary:**

## Paper Topic And Main Contributions:

The paper proposes a new Visual Question Answering for early diagnosis of preschool children to explore the visual behaviours of ASD children. Additionally, it proposes the VQA-guided cooperative ASD screening network (VQA-CASN) to explore the eye tracking paradigm. The paper aims to identify preschool children with ASD by focusing on proposed eye-tracking paradigm, new dataset and model (by capturing eye movement features in a task-agnostic manner and differences in the regions of interest).

Eye tracking paradigm empowered by VQA4ASD dataset and VQA-guided cooperative ASD screening network (VQA-CASN) can be mentioned as the main contributions of the paper.

**Strengths:**

## Strengths:

1. In medical application level, the usage of gaze feature for early diagnosis of preschool children is new.

2. In idea level, VQA-CASN sounds really promising. Additionally, I believe that this idea can be explored more in different applications.

**Limitations:**

## Weakness:

1. Generally, the main purpose of using the human gaze feature is to capture the relevant objects or actions to support relevant hypotheses. Therefore, this approach is not novel enough for the scientific level.

2. According to the references cited by the authors, this dataset cannot be considered novel either. However, a new dataset has been explored for younger subjects (as compared in the paper). Additionally, a Gaze features-enhanced dataset has already been explored for eye-tracking paradigms in different applications before. Please, re-explore the literature.

3. The VQA dataset is one of the priorities of this paper, and there is no detailed explanation about the questions.

## Questions For The Authors:

This is an important task, and the paper focuses on preschool ASD children. Questions in the VQA dataset are also crucial. Eye tracking is a paradigm followed, and the proposed hypothesis is tested with the model and results. However, questions are key points. I believe that you have already been careful about them, but I would like to see some clarification in the paper. I felt unconvinced about the questions and preparations.

1. What are the professions (pedagogues, medical doctors, teachers, etc.) of people who designed the questions?

2. Were the questions prepared manually or AI-generated?

3. Were the questions designed inspired by pedagogical guidance?

4. What are the specific reasons for preparing the proposed questions (in the paper and appendix)?

5. How do these questions meet your expectations?

6. What are the protocols or fundamental rules for preparing the questions?

7. How can you prove these questions are easy tasks for 2-6-year-old children?

8. Please provide some statistical information about the subjects and experiment environments.

9. How big were the figures shown to the subjects? (I think that affects children's saccades and fixations.)

10. Additionally, how many 2 or 3-year-old children joined the experiments? I ask this question because some questions in the supplementary seem really hard for them. Because some questions consist of “How many…”. Therefore, a questions rises: How can 2 or 3-year-old children give the right answer? Even if you do not expect them to give the right answer, and you expect them to check the right objects, those children can be confused while checking the correct object and numbers. Consequently, what are the prediction results of your model in such cases?

11. Please provide some statistical data about VQA4ASD in terms of the number of questions, word cloud distributions, etc.

12. If the new questions are collected with new figures and scenes, why were only questions and answers prepared on Saliency4ASD? In short, what is the key point for collecting a new dataset with new figures and scenes?

13. Isn’t it enough to curate only new QA pairs relevant to the Saliency4ASD dataset? If the only difference between the new dataset and Saliency4ASD is questions, then the preparation of that new dataset may not convince the community about why we need a new dataset.

## Typos Grammar Style And Presentation Improvements:

1. This paper is well organized and written.

**Suitability:**

3

---

### Official Review · Reviewer_vFoR · 2024-05-22

**Rating:** 4
**Confidence:** 3

**Summary:**

The authors create a VQA-driven eye-tracking dataset for early diagnosis of ASD in children with two goals: (1) to reduce intra-variance in the control group by providing explicit guidance; (2) to reveal social and communication skills, particularly the ability to receive questions and respond, indicating social and attentional deficits. In addition, they develop a VQA-guided cooperative ASD screening network (VQA-CASN), in which both task-agnostic and task-specific visual scan paths are explored simultaneously for ASD screening. The results show the effectiveness of their proposed VQA paradigm and the VQA-CASN model.

**Strengths:**

This work is interesting and well-written.

**Limitations:**

Comments 1: This dataset is generated by 2-6-year-old children. Does it appear that the age factor affects the understanding of the question. It is suggested that the authors add the performance of model at different age stages.

Comments 2: In the Results, evaluation metrics (e.g., sensitive (Sen) and accuracy (Acc)) are not written in a harmonized way, i.e., sometimes it's a full name, sometimes it's an abbreviation.

Comments 3: For the VQA-CASN, it is suggested that the authors consider more novel feature fusion methods to replace the concatenation.

**Suitability:**

3

---

### Official Review · Reviewer_YwbR · 2024-05-25

**Rating:** 4
**Confidence:** 3

**Summary:**

This paper conducts a study that uses VQA-driven eye-tracking techniques for ASD diagnosis.

**Strengths:**

1. To the best of our knowledge, this seems the first work studies VQA based eye movement for ASD.
2. Due to the disease of ASD, it seems significant to conduct VQA base eye tracking for ASD.

**Limitations:**

1. The age ranges from 2-6, it is hard to let them understand the question, even for high functioning ASD and env for normal 2 years old children. This is the most important thing in the experiment, and should be seriously answered.
2. The dataset size seems small, only 25+25 images were included.

**Suitability:**

2

---

### Meta-Review · Area_Chair_nSkP · 2024-07-07

**Recommendation:** Accept (Poster)
**Confidence:** 4

**Metareview:**

In the first round of reviews, this paper got 2 borderline acceptances and 1 borderline rejection. The authors submitted a rebuttal, which satisfied two of the reviewers. Taking into account the reviews, the rebuttal, the final comments by the reviewers after the rebuttal and the interesting and innovative topic addressed in the paper, I lean towards recommending this paper for acceptance as poster.